# Association of Household Income Level with Vitamin and Mineral Intake

**DOI:** 10.3390/nu14010038

**Published:** 2021-12-23

**Authors:** Haegyu Oh, Juyeon Kim, Yune Huh, Seung Hoon Kim, Sung-In Jang

**Affiliations:** 1Premedical Courses, Yonsei University College of Medicine, Seoul 03722, Korea; studyinghuman@yonsei.ac.kr (H.O.); kjyeon0515@yonsei.ac.kr (J.K.); twinring77@yonsei.ac.kr (Y.H.); 2Department of Preventive Medicine, Yonsei University College of Medicine, Seoul 03722, Korea; 3Institute of Health Services Research, Yonsei University, Seoul 03722, Korea

**Keywords:** nutrient intake, South Korea, minerals, vitamins

## Abstract

Appropriate nutrient intake is essential for maintaining health and resisting disease. The current study investigated the association between household income quintile and nutrient intake using data from KNHANES 2019. A total of 5088 South Korean adults were analyzed. The estimated average requirement cut-point method, extended to handle participants with intakes higher than the tolerable upper level, was utilized to determine the need for dietary modification. The suitability of overall vitamin, overall mineral, and individual nutrient intake was evaluated by logistic regression analysis. Subgroup analysis was performed on overall mineral intake suitability. None of the individual nutrients had an intake ratio of over 70%, with the ratio being under 30% for some nutrients. The intake of iron, phosphorus, vitamin B9, and vitamin C had a significant upward trend as household income rose. A subgroup analysis revealed sex differences in the trends of overall mineral intake. The results revealed that some nutrients are not consumed appropriately in the Korean population. Furthermore, they suggest that household income is significantly associated with the intake of overall minerals and several individual nutrients. These results suggest that nutritional assistance is required for certain vulnerable groups, and provide supplementary data for appropriate interventions or further research.

## 1. Introduction

The average energy intake per day has grown in both the USA [1] and South Korea [2], but this does not necessarily mean that nutritional balance has also improved. To achieve a balanced level of nutrition, one needs to consume not only sufficient energy but also proper amounts of vitamins and minerals. Deficiencies of any vitamin or mineral are implicated in various chronic diseases [3]. For instance, deficiency of vitamin B may cause beriberi [4], anemia [5], pellagra [6], spinal cord lesions (e.g., spina bifida) and peripheral neuropathy [7]. Severe conditions including osteoporosis, anemia, and peripheral neuropathy can also occur if insufficient calcium, phosphorus, and iron are consumed [7,8,9,10,11]. Sufficient vitamin and mineral intake can be achieved through a balanced diet [12].

Household income may have a significant effect on one’s diet, which could lead to differences in nutrient intake. Compared to the low-income class, the wealthy are more likely to have a diet with high-quality ingredients, and a better understanding of the importance of a balanced diet [13]. According to a 2001 study on Americans and Canadians, social standing, specifically, household income and education and working status, have an impact on nutrition intake [14]. There might also be a similar tendency in South Korea. According to 2021 OECD statistics, South Korea ranked fourth on the relative poverty scale, defined by the proportion of people whose income is less than 50% of the standard median income [15]. This wide income gap may be reflected in their nutrient intake.

Several previous studies in South Korea have investigated the association between income status and the intake of nutrients in South Korean adults. An analysis from the 2011 Korean National Health and Nutrition Examination Survey (KNHANES) showed that the average intake of energy and all nutrients increased in higher household income quartiles [16]. However, there are not enough recent studies focusing on vitamin and mineral intake in South Korea. Therefore, the current study examined the correlation between household income quintile and vitamin or mineral intake. Data from the 2019 KNHANES were used. The intake of overall vitamins and minerals, and of individual nutrients, was examined.

## 2. Materials and Methods

### 2.1. Data Collection

In the current study, data were extracted from the 2019 KNHANES Ⅷ-1, a cross-sectional sample survey conducted by the Korea Disease Control and Prevention Agency [17]. The overall participation rate was 74.7%. The questionnaire investigated health behavior, food and nutrition intake, and prevalence of chronic diseases in the Korean population, along with fundamental information such as age, sex, and type of inhabitation. The questionnaire consisted of three parts: a health interview survey, a health examination survey, and a nutrition survey. Among these, the nutrition survey collected data about dietary behavior, dietary supplements, nutritional knowledge, food stability, and daily food intake. Food intake was examined by the 24-h dietary recall method, with the participant recalling what they had eaten in the last 24 h. We analyzed data from the nutrition survey, focusing on the intake of each vitamin and mineral. The study was conducted only on adults in order to take alcohol consumption and smoking into account as confounding variables. The number of total participants was 8110, but only 6606 participants were above the age of 18. Of these, 5088 had no missing data for the variables investigated in this study. The size of the sample was large enough to draw generalized conclusions about the whole South Korean population. The process of participant recruitment is shown in Figure 1.

### 2.2. Variables

The variable of interest in this study was household income quintile, with quintiles 1 to 5 indicating the lowest to the highest income group. Income was obtained as the monthly household income divided by the square root of the number of household members, for every member, including those without economic activity. Quintiles were calculated for each group, divided by age and gender. Suitability of intake indicated whether the intake of each target nutrient was adequate; 0 stood for unsuitable, and 1 for suitable. Overall intake for vitamins and minerals was considered suitable when the intake of every vitamin or mineral examined was adequate. To establish an objective threshold for the intake suitability of each target nutrient, a revised version of the Estimated Average Requirement (EAR) cut-point method was chosen. EAR defines the level of nutrition that meets the daily needs of 50% of healthy individuals within a target group, and is obtained as the median of the requirement values [12]. The EAR cut-point method is widely used in nutrition research because it enables simple estimations to be made of the proportion of subjects consuming nutrients below required levels [18]. In a similar manner, nutrient intake above the tolerable upper intake level (UL) was considered as unsuitable. UL is the maximum intake which is unlikely to cause adverse effects, and is determined by dose-response assessments. Though using UL as a cut-point to assess nutrient intake is not recommended [19,20], it was used as a reference, since it was the only value available from the Dietary Reference Intakes for Koreans (KDRIs) EAR and UL values were both taken from the 2020 KDRIs [21]. Among all vitamins and minerals, vitamins A, B1/B2/B3/B9, and C, as well as calcium, phosphorus, and iron, whose EARs are shown in KDRIs and intake amounts are available in KNHANES, were selected for analysis. UL values could not be found for vitamins B1 and B2, so these nutrients were analyzed only with EAR values. We also considered confounding covariates such as sex, age (19–29, 30–39, 40–49, 50–59, 60–69, and ≥70), educational level (below middle school, high school, college, and graduate school), how many chronic diseases the participants have (0, 1, and ≥2), frequency of alcohol drinking (never, 2–4 times/month, and 2–4 times/week) and smoking (never, quit smoking, <1 time/day, and ≥1 time/day), obesity status (underweight, normal, overweight, and obese), nutritional education status (whether or not they had received nutrition education and counseling conducted at public health centers, ward offices, community centers, welfare facilities, schools, or hospitals in the past year—yes or no), dietary control (whether or not to control overall diet for reasons such as disease, weight control—yes or no), and energy intake (kcal).

### 2.3. Statistical Analysis

To analyze the associations between nutrient intake suitability and each variable, chi-square and one-way analysis of variance tests were utilized for categorical and continuous variables, respectively. Means and standard deviations were calculated for continuous variables. Associations between intake suitability and each variable were then examined by logistic regression analysis. Subgroup analysis of overall mineral intake was also performed by logistic regression analysis. Adjusted odds ratios (AORs) and 95% confidence intervals (CIs) were evaluated. The entire process was carried out with the SAS software, version 9.4 (SAS Institute, Cary, NC, USA). A weighted logistic regression analysis was performed to account for the complex and stratified sampling design, and a two-sided *p*-value < 0.05 indicated statistical significance.

## 3. Results

### 3.1. Analysis of Overall Intake Suitability

The general characteristics of the KNHANES 2019 study participants according to overall vitamin or mineral intake are shown in Table 1. A total of 5088 participants above the age of 18 were analyzed. Only 441 participants (8.7%) had suitable overall vitamin intake, while 1210 participants (23.8%) had suitable overall mineral intake. The proportion of participants with suitable vitamin or mineral intake significantly increased as household income rose.

Table 2 shows the results of the logistic regression analysis performed to examine the correlation between household income quintile and overall vitamin or mineral intake. Though no significant trend was found for either vitamin or mineral intake suitability, the mineral intake suitability of quintiles 3 and 5 showed a significant disparity compared to quintile 1. No such significant difference was found for vitamin intake suitability.

### 3.2. Analysis of Individual Nutrient Intake Suitability

The intake suitability of each individual nutrient was then examined. Appendix A shows the intake suitability of nutrients according to household income quintile. Less than half of the participants consumed adequate amounts of vitamin A, vitamin B3, vitamin C, and calcium. For every nutrient except iron, the percentage of participants with suitable intake had an increasing tendency according to household income.

A logistic regression analysis was performed to examine the association between household income and the intake suitability of individual nutrients. Analyses were performed similarly to the analysis performed in Table 2, and only the results according to household income are shown in Table 3. Quintile 5 showed a significant disparity compared to quintile 1 for vitamins B3, B9, and C, as well as iron, calcium, and phosphorus. Still, only the trends for vitamin B9, vitamin C, phosphorus, and iron were significant. On the other hand, the associations between vitamins B2 and B3, calcium, iron, and phosphorus intake suitability and household income quintile were not significant.

### 3.3. Subgroup Analysis of Overall Mineral Intake Suitability

As there was generally a greater association between household income quintile and the intake suitability of minerals relative to vitamins, a subgroup analysis was conducted on overall mineral intake suitability; the results are shown in Table 4. Household income quintile had a significant association with mineral intake suitability only in women, with all the AORs of quintiles 2 to 5 being greater than 1.7. The AOR values for participants with below middle school education, with one or more chronic diseases, who drank less than once a week, who never smoked, with no nutritional education, and who were not on a diet, showed a similar trend.

## 4. Discussion

Household income quintile group 5 was significantly more likely to exhibit suitable overall mineral consumption compared to group 1. On the other hand, overall vitamin suitability did not have a significant association with household income quintile. Further investigation on individual nutrients revealed that higher income households tended to show a higher percentage of suitable vitamin/mineral consumption. Iron intake showed a significantly increasing trend, along with phosphorus, vitamin B9, and vitamin C. Therefore, we concluded that the intake suitability of certain vitamins and minerals, as well as overall minerals, is associated with household income.

Only overall mineral intake suitability was significantly associated with household income, as opposed to overall vitamin suitability. One potential explanation for this is that more vitamins were examined than minerals, and hence, overall vitamin suitability was subject to stricter conditions than overall mineral suitability. In addition, the intake suitability of certain nutrients was significantly associated with household income. A number of factors may influence this phenomenon. First, financial strain may limit one’s food purchases. In the USA and Australia, lower income groups tend to purchase more energy-dense foods with added sugar and fat [22,23]. Furthermore, we believe that the intake suitability of nutrients which are fortified in affordable and frequently eaten foods, such as rice, will be less associated with income. Second, a lack of income may lead to a lack of time to engage in healthy eating practices. In the USA, the low household income tertile group tended to prefer food that requires less preparation time [24]. Therefore, they were more exposed to fast food and tended to suffer from negative health outcomes such as obesity and acute coronary syndromes [25]. Further research is required to elucidate whether and why income level influences different nutrients differently.

The present study has several implications. First, to our knowledge, this is the first study to establish an association between nutritional health concerns and income level in South Korea. Several studies have tried to identify the association between income and nutrient intake in South Korean citizens. However, such studies only identified discrepancies in nutrient intake and food groups across different income levels [26]. They did not conduct detailed analyses of whether said differences lead to actual health issues. Our study adopted the EAR cut-point method and discovered that income level also influences nutrient consumption according to dietary reference intakes (DRIs). As nutrient consumption below the EAR or over the UL implies elevated likelihood of health issues, we could establish a standard with which to quantify the probability of health issues. In addition, the percentages of individuals consuming a suitable amount of vitamin B1 (68.3%) and vitamin B2 (66.8%) were consistent with previous reports [27,28] in South Korean adults. These results were determined based on urinary vitamins, which further consolidates the reliability of using DRI based methods for analyzing nutrient intake.

Second, the effects of confounding factors such as smoking status, drinking status, nutritional education status, and calorie intake were excluded using logistic regression analysis. Such a method has not been used in any study to date on South Korean participants.

Last, our study was conducted based on a nationwide survey. Therefore, our sample had an adequate representation of participants to draw conclusions which are applicable to the entire South Korean population. A weighted logistic regression analysis was used to eliminate errors that may have occurred from the sampling process, including nonresponse or nonparticipation biases.

However, the present study also has several limitations. First, it could not confirm a causal relationship between income and nutrient consumption, as it was based on cross-sectional data. A longitudinal analysis should be performed to firmly establish causality.

Second, our study may have been affected by inherent limitations of the 24 h dietary recall method, which is known to be intrinsically biased since it is memory dependent [29]. It is known that over- and under-reporting of nutrient intake frequently occur when data are collected with questionnaires [30,31]. Estimations of nutrition intake based on questionnaire responses have shown differences compared with their corresponding biomarkers in blood or urine samples collected through laboratory means, due to individual differences of metabolism [32]. The present study would have implemented this approach if the data had also included biomarkers measured in the biological samples of each participant. Nevertheless, we found a general consistency in the results on the intake suitability of vitamins B1 and B2, which suggests that errors caused by the 24 h dietary recall method may be insignificant.

Third, the results of our study were confined for nine nutrients, since KDRIs do not provide EAR cut-points for other vitamins and minerals. Alternative nutrient profiling scoring methods, such as the Korean Healthy Eating Index [33], should be applied along with the current method to measure the suitability of other nutrient levels.

Fourth, DRI methods can be biased by within-individual variance [34]. Notably, DRI methods tend to lead to biased estimates, as supplementary consumption sources can compensate for daily nutrient intake from regular food sources. The data we used could not sufficiently measure supplement consumption, as the frequency of vitamin and mineral supplement consumption was not included in the questionnaire. More detailed sampling methods covering the amount and frequency of supplement consumption should be used to accurately capture the within-individual variance of nutrient intake.

Interestingly, educational level was found to significantly influence overall mineral intake. In the USA, higher education level is associated with higher income, with family financial income increasing up to fivefold based on family education [35]. In South Korea, is has been reported that the consumption of mineral- and vitamin-rich fruits and vegetables correlated with education levels [36]. Our findings may further consolidate the assumption that higher income, better education, and better nutrient consumption may be correlated with one another.

Furthermore, our subgroup analysis showed a difference in the trends of overall mineral intake between men and women. Women had a significantly increasing trend between income quintile and overall mineral intake, while no such association was observed for men. In South Korea, the impact of income and education on morbidity and self-reported health status is more pronounced in women than in men [37]. This result may suggest that women in the first income quintile are especially exposed to nutrient deficient diets. Further subgroup analyses may reveal groups that have particularly unhealthy dietary habits.

## 5. Conclusions

The present study revealed that households with higher income had more suitable dietary habits in terms of certain mineral or vitamin consumption. Unlike previous studies, we associated socioeconomic status with actual health risks using the EAR cut-point method. Moreover, we discovered that nutrient consumption among women was more drastically affected by income than among men. Our study suggests that more guidance and support are needed for groups that display especially low nutrient intake. Additional research is required to identify vulnerabilities in nutrition from a demographic perspective, and to provide appropriate support.

## Figures and Tables

**Figure 1 nutrients-14-00038-f001:**
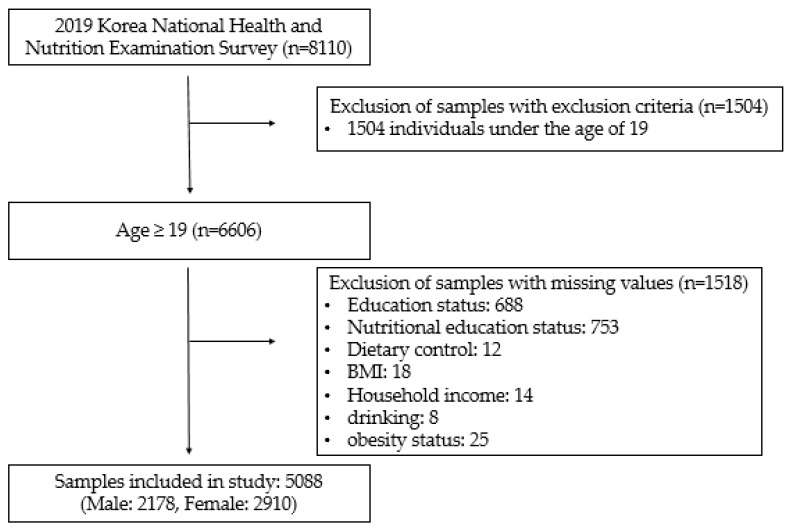
A schematic diagram showing the process of participant recruitment. Individuals under the age of 19 or with missing values were excluded. BMI: Body Mass Index.

**Table 1 nutrients-14-00038-t001:** General characteristics of study subjects according to overall vitamin, mineral intake. The number and percentage of participants from each group, and the *p*-values for each criterion, are shown. BMI: Body Mass Index.

Variables			Overall Vitamins					Overall Minerals				
	Total		Suitability = 0		Suitability = 1		*p*-Value	Suitability = 0		Suitability = 1		*p*-Value
	N	%	N	%	N	%		N	%	N	%	
Total (*n* = 5088)	5088	100	4647	91.3	441	8.7		3878	76.2	1210	23.8	
Household income quintile							0.004					<0.0001
1 (very low)	765	15.0	714	93.3	51	6.7		643	84.1	122	15.9	
2 (low)	946	18.6	879	92.9	67	7.1		740	78.2	206	21.8	
3 (medium)	1001	19.7	917	91.6	84	8.4		750	74.9	251	25.1	
4 (high)	1159	22.8	1054	90.9	105	9.1		871	75.2	288	24.8	
5 (very high)	1217	23.9	1083	89.0	134	11.0		874	71.8	343	28.2	
Sex							0.823					<0.0001
Male	2178	42.8	1987	91.2	191	8.8		1545	70.9	633	29.1	
Female	2910	57.2	2660	91.4	250	8.6		2333	80.2	577	19.8	
Age (years)							0.003					<0.0001
19~29	579	11.4	541	93.4	38	6.6		469	81.0	119	20.6	
30~39	737	14.5	675	91.6	62	8.4		572	77.6	165	22.4	
40~49	936	18.4	847	90.5	89	9.5		724	77.4	212	22.6	
50~59	961	18.9	857	89.2	104	10.8		675	70.2	286	29.8	
60~69	928	18.2	839	90.4	89	9.6		673	72.5	255	27.5	
70+	947	18.6	888	93.8	59	6.2		774	81.7	173	18.3	
Educational level							0.002					<0.0001
below middle school	1360	26.7	1275	93.8	85	6.3		1120	82.4	240	17.6	
high school	1407	27.7	1276	90.7	131	9.3		1049	74.6	358	25.4	
College	2064	40.6	1868	90.5	196	9.5		1534	74.3	530	25.7	
graduate school	257	5.1	228	88.7	29	11.3		175	68.1	82	31.9	
Chronic disease ^a^							0.103					0.152
0	3354	65.9	3043	90.7	311	9.3		2535	75.6	819	24.4	
1	931	18.3	861	92.5	70	7.5		710	76.3	221	23.7	
≥2	803	15.8	743	92.5	60	7.5		633	78.8	170	21.2	
Alcohol							0.167					0.013
Never	559	11.0	516	92.3	43	7.7		453	81.0	106	19.0	
Occasionally, 2~4 times/month	3489	68.6	3169	90.8	320	9.2		2648	75.9	841	24.1	
2~4 times/week	1040	20.4	962	92.5	78	7.5		777	74.7	263	25.3	
Smoking							0.670					<0.0001
Never	3088	60.7	2813	91.1	275	8.9		2430	78.7	658	21.3	
Quit smoking	1167	22.9	1065	91.3	102	8.7		834	71.5	333	28.5	
<1 time/day	125	2.5	117	93.6	8	6.4		100	80.0	25	20.0	
≥1 time/day	708	13.9	652	92.1	56	7.9		514	72.6	194	27.4	
Obesity Status ^b^							0.217					0.335
Underweight	198	3.9	183	92.4	15	7.6		159	80.3	39	19.7	
Normal	2001	39.3	1807	90.3	194	9.7		1519	75.9	482	24.1	
Overweight	1180	23.2	1085	91.9	95	8.1		885	75.0	295	25.0	
Obese	1709	33.6	1572	92.0	137	8.0		1315	76.9	394	23.1	
Nutritional Education Status							0.000					0.479
Yes	278	5.5	237	85.3	41	14.7		207	74.5	71	25.5	
No	4810	94.5	4410	91.7	400	8.3		3671	76.3	1139	23.7	
Diet							0.057					0.372
Yes	1473	29.0	1328	90.2	145	9.8		1135	77.1	338	22.9	
No	3615	71.0	3319	91.8	296	8.2		2743	75.9	872	24.1	
Energy intake (Mean, SD, kcal)	1849	862.3	1771	802.6	2667	1030	<0.0001	1663	750.5	2255	924.3	<0.0001

^a^ Defined as the sum of the number of chronic diseases: hypertension, dyslipidemia, stroke, myocardial infarction, angina pectoris, and diabetes mellitus. ^b^ BMI values were used to determine obesity status. 0 < BMI < 18.5: underweight, 18.5 ≤ BMI < 23.0: normal, 23.0 ≤ BMI < 25.0: overweight, BMI ≥ 25.0: obese.

**Table 2 nutrients-14-00038-t002:** Logistic regression analysis of factors related to vitamin, mineral intake. The groups at the top of each criterion were compared with each of the other groups. Adjusted odds ratio (AOR), 95% confidence intervals (CI), and *p*-values for trends are shown. BMI: Body Mass Index.

Variables	Overall Vitamins					Overall Minerals				
	AOR	95% CI			*p*-Value	AOR	95% CI			*p*-Value
Household income quintile					0.165					0.129
1 (very low)	**1.00**					**1.00**				
2 (low)	0.89	0.51	–	1.53		1.42	1.00	–	2.01	
3 (medium)	0.95	0.53	–	1.70		1.41	1.01	–	1.96	
4 (high)	1.07	0.63	–	1.82		1.30	0.90	–	1.86	
5 (very high)	1.21	0.69	–	2.10		1.54	1.07	–	2.21	
Sex										
Male	**1.00**					**1.00**				
Female	1.65	1.11	–	2.44		1.10	0.85	–	1.44	
Age (years)										
19~29	**1.00**					**1.00**				
30~39	1.80	1.00	–	3.24		1.18	0.81	–	1.71	
40~49	2.48	1.46	–	4.19		1.45	1.01	–	2.06	
50~59	3.33	1.88	–	5.90		2.87	2.00	–	4.12	
60~69	4.99	2.76	–	9.04		3.94	2.58	–	6.00	
70+	5.23	2.54	–	10.78		3.73	2.32	–	5.98	
Educational level										
below middle school	**1.00**					**1.00**				
high school	1.59	0.97	–	2.61		1.56	1.16	–	2.09	
college	1.61	0.98	–	2.66		1.71	1.22	–	2.39	
graduate school	1.74	0.98	–	3.08		1.88	1.21	–	2.93	
Chronic disease ^a^										
0	**1.00**					**1.00**				
1	0.95	0.66	–	1.36		1.03	0.82	–	1.30	
≥2	0.79	0.53	–	1.18		0.86	0.65	–	1.13	
Alcohol										
Never	1.00					1.00				
Occasionally, 2~4 times/month	0.97	0.64	–	1.46		1.12	0.81	–	1.54	
2~4 times/week	0.41	0.24	–	0.72		0.71	0.48	–	1.05	
Smoking										
Never	**1.00**					**1.00**				
Quit smoking	0.90	0.57	–	1.43		0.98	0.73	–	1.31	
<1 time/day	1.02	0.33	–	3.18		0.79	0.42	–	1.49	
≥1 time/day	0.76	0.47	–	1.23		0.90	0.65	–	1.24	
Obesity Status ^b^										
Normal	**1.00**					**1.00**				
Underweight	1.24	0.69	–	2.23		1.13	0.75	–	1.69	
Overweight	0.73	0.54	–	0.99		0.90	0.73	–	1.10	
Obese	0.80	0.59	–	1.09		0.85	0.71	–	1.02	
Nutritional Education Status										
No	**1.00**					**1.00**				
Yes	1.85	1.19	–	2.88		1.14	0.79	–	1.65	
Diet										
No	**1.00**					**1.00**				
Yes	1.48	1.14	–	1.92		1.10	0.91	–	1.33	
Energy intake (kcal)	1.00	1.00	–	1.00		1.00	1.00	–	1.00	

^a^ Defined as the sum of the number of chronic diseases: hypertension, dyslipidemia, stroke, myocardial infarction, angina pectoris, and diabetes mellitus. ^b^ BMI values were used to determine obesity status. 0 < BMI < 18.5: underweight, 18.5 ≤ BMI < 23.0: normal, 23.0 ≤ BMI < 25.0: overweight, BMI ≥ 25.0: obese.

**Table 3 nutrients-14-00038-t003:** Logistic regression analysis results of factors related to individual vitamin or mineral intake. Only the results according to household income quintile are shown. Adjusted odds ratio (AOR)s, 95% confidence intervals (CI), and trend *p*-values are shown.

	Household Income Quintile																	
Outcomes	Quintile 1 (Very Low)	Quintile 2 (Low)				Quintile 3 (Medium)				Quintile 4 (High)				Quintile 5 (Very High)				*p*-Value
	AOR	AOR	95% CI			AOR	95% CI			AOR	95% CI			AOR	95% CI			
Vitamin A intake suitability	1.00	0.89	0.51	–	1.53	0.95	0.53	–	1.70	1.07	0.63	–	1.82	1.21	0.69	–	2.10	0.212
Vitamin B1 intake suitability	1.00	1.03	0.74	–	1.43	1.05	0.51	–	1.53	1.10	0.77	–	1.57	1.20	0.82	–	1.77	0.762
Vitamin B2 intake suitability	1.00	0.93	0.68	–	1.26	0.97	0.71	–	1.34	0.91	0.66	–	1.26	0.94	0.67	–	1.32	0.216
Vitamin B3 intake suitability	1.00	1.28	0.98	–	1.66	1.09	0.78	–	1.52	1.11	0.80	–	1.53	1.44	1.05	–	1.98	0.103
Vitamin B9 intake suitability	1.00	1.03	0.76	–	1.38	1.21	0.85	–	1.71	1.26	0.90	–	1.77	1.44	1.03	–	2.03	0.005
Vitamin C intake suitability	1.00	0.97	0.71	–	1.33	1.24	0.89	–	1.72	1.25	0.91	–	1.71	1.47	1.03	–	2.09	0.006
Ca intake suitability	1.00	1.43	1.02	–	2.01	1.55	1.12	–	2.13	1.26	0.90	–	1.75	1.40	1.00	–	1.96	0.530
P intake suitability	1.00	1.04	0.75	–	1.43	1.00	0.70	–	1.43	1.20	0.85	–	1.71	1.67	1.19	–	2.34	0.011
Fe intake suitability	1.00	1.51	1.06	–	2.13	1.49	1.06	–	2.10	1.55	1.06	–	2.27	1.84	1.23	–	2.75	<0.001

**Table 4 nutrients-14-00038-t004:** Subgroup analysis of overall mineral intake suitability by logistic regression analysis. For each subgroup, every criterion shown in the table below except for the one corresponding to the subgroup was analyzed. Each quintile group was compared with quintile 1. Adjusted odds ratio (AOR)s, 95% confidence intervals (CI), and trend *p*-values are shown. BMI: Body Mass Index.

Variables	Household Income Quintile																	
	Quintile 1 (Very Low)	Quintile 2 (Low)				Quintile 3 (Medium)				Quintile 4 (High)				Quintile 5 (Very High)				*p*-Value
	AOR	AOR	95% CI			AOR	95% CI			AOR	95% CI			AOR	95% CI			
Sex																		
Male	1.00	1.08	0.70	–	1.68	1.18	0.78	–	1.79	0.92	0.58	–	1.47	1.17	0.76	-	1.81	0.767
Female	1.00	1.79	1.04	–	3.07	1.74	1.01	–	3.01	1.90	1.11	–	3.23	2.12	1.16	-	3.88	0.041
Age (years)												–						
19~29	1.00	1.89	0.50	–	7.18	1.21	0.43	–	3.41	1.07	0.35	–	3.31	1.89	0.65	-	5.46	0.295
30~39	1.00	1.41	0.48	–	4.13	1.04	0.33	–	3.25	1.17	0.35	–	3.89	0.89	0.27	-	2.97	0.354
40~49	1.00	0.76	0.24	–	2.38	0.67	0.23	–	1.94	0.80	0.27	–	2.36	0.85	0.30	-	2.43	0.704
50~59	1.00	0.99	0.39	–	2.51	1.03	0.42	–	2.54	1.14	0.50	–	2.64	1.48	0.67	–	3.25	0.085
60~69	1.00	1.15	0.54	–	2.48	2.65	1.22	–	5.75	1.34	0.63	–	2.86	1.62	0.71	–	3.65	0.238
70+	1.00	1.77	1.06	–	2.96	1.67	0.93	–	3.01	0.90	0.47	–	1.72	0.77	0.20	–	3.01	0.995
Educational level																		
below middle school	1.00	1.50	0.89	–	2.53	1.98	1.15	–	3.38	1.62	0.88	–	2.98	1.72	0.78	–	3.78	0.061
high school	1.00	1.46	0.75	–	2.84	0.98	0.54	–	1.78	0.98	0.52	–	1.83	1.10	0.60	–	2.02	0.481
college	1.00	1.05	0.54	–	2.04	1.22	0.68	–	2.19	1.02	0.52	–	1.99	1.38	0.74	–	2.59	0.224
graduate school	1.00	0.77	0.06	–	10.22	1.20	0.13	–	11.26	1.84	0.20	–	17.21	1.28	0.14	–	12.07	0.726
Chronic disease ^a^																		
0.000	1.00	1.15	0.73	–	1.81	1.04	0.66	–	1.62	1.10	0.69	–	1.76	1.23	0.77	–	1.97	0.384
1.000	1.00	1.87	0.96	–	3.66	3.25	1.72	–	6.12	1.54	0.73	–	3.21	2.57	1.25	–	5.27	0.084
≥2	1.00	1.68	0.80	–	3.53	1.65	0.77	–	3.51	1.44	0.55	–	3.79	1.61	0.69	–	3.76	0.517
Alcohol																		
Never	1.00	1.79	0.69	–	4.62	4.12	1.57	–	10.84	1.85	0.69	–	4.98	1.55	0.46	–	5.19	0.307
Occasionally, 2~4 times/month	1.00	1.56	1.04	–	2.34	1.35	0.88	–	2.07	1.37	0.90	–	2.09	1.64	1.11	–	2.44	0.099
2~4 times/week	1.00	1.03	0.49	–	2.17	1.03	0.48	–	2.20	0.88	0.39	–	1.98	1.06	0.47	–	2.40	0.981
Smoking																		
Never	1.00	1.76	1.08	–	2.87	1.48	0.89	–	2.45	1.47	0.90	–	2.40	1.66	0.98	–	2.80	0.330
Quit smoking	1.00	1.24	0.70	–	2.20	1.40	0.77	–	2.53	1.22	0.65	–	2.28	1.55	0.90	–	2.65	0.230
<1 time/day	1.00	0.42	0.02	–	8.02	12.23	1.65	–	90.56	4.95	0.22	–	112.47	13.46	1.34	–	135.04	0.065
≥1 time/day	1.00	0.88	0.42	–	1.84	1.13	0.52	–	2.45	0.88	0.39	–	1.97	0.98	0.41	–	2.38	0.958
Obesity Status ^b^																		
Normal	1.00	1.68	0.85	–	3.33	1.19	0.63	–	2.28	1.74	0.92	–	3.29	1.98	0.98	–	4.00	0.052
Underweight	1.00	0.28	0.05	–	1.73	1.09	0.21	–	5.65	1.80	0.24	–	13.41	1.62	0.28	–	9.45	0.247
Overweight	1.00	1.47	0.81	–	2.66	1.34	0.74	–	2.43	1.14	0.62	–	2.12	1.25	0.70	–	2.23	0.910
Obese	1.00	1.06	0.61	–	1.83	1.63	0.96	–	2.76	0.97	0.54	–	1.76	1.22	0.70	–	2.13	0.970
Nutritional Education Status																		
No	1.00	1.47	1.02	–	2.11	1.50	1.07	–	2.11	1.39	0.96	–	2.00	1.59	1.09	–	2.30	0.124
Yes	1.00	0.74	0.21	–	2.58	0.52	0.15	–	1.83	0.42	0.07	–	2.63	1.32	0.27	–	6.51	0.526
Diet																		
No	1.00	1.66	1.09	–	2.52	1.46	0.96	–	2.21	1.56	1.02	–	2.39	1.63	1.05	–	2.53	0.169
Yes	1.00	0.98	0.52	–	1.83	1.38	0.76	–	2.53	0.86	0.46	–	1.60	1.42	0.79	–	2.56	0.297

^a^ Defined as the sum of the number of chronic diseases: hypertension, dyslipidemia, stroke, myocardial infarction, angina pectoris, and diabetes mellitus. *^b^* BMI values were used to determine obesity status. 0 < BMI < 18.5: underweight, 18.5 ≤ BMI < 23.0: normal, 23.0 ≤ BMI < 25.0: overweight, BMI ≥ 25.0: obese.

## Data Availability

Publicly available datasets were analyzed in this study. The data can be found in the official website of the Korea Disease Control and Prevention Agency (https://knhanes.kdca.go.kr/knhanes/sub03/sub03_02_05.do, accessed on 17 November 2021).

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
