# Peer review of "Association of Household Income Level with Vitamin and Mineral Intake"

_nutrients, 2021, doi:10.3390/nu14010038_

Round 1

Reviewer 1 Report

  1. The phrase "nutrient uptake" and "uptake suitability" is used several times within this manuscript, but the use of the word uptake is improper for a study based on nutritional records since absorption is not measured. Intake (or similar terminology) should be used exclusively.
  2. In the Introduction, the 2021 OECD statistics should be cited.
  3. The variable "suitability of nutrition uptake" is somewhat vaguely defined in the Materials and Methods. Is this indicating whether intake met or exceeded the EAR for a particular micronutrient or all micronutrients combined? For example, if a score of 1 is assigned, does this indicate the intake of all micronutrients exceeded the EAR? This should be stated plainly within the Methods section.
  4. It is unclear why only certain vitamins and minerals were analyzed and not all vitamins and minerals. 
  5. Appropriate significant digits should be used when displaying p-values and confidence intervals. The use of four significant digits is excessive and wholly uninformative. 
  6. Please explain the variables "nutrition educational status" and "dietary control"
  7. In Table 2, why were the statistics for the other regression variables (age, weight, etc.) not performed?
  8. Similar to question 3, it is unclear what is being reported in Table 4. Is this a synthesis of individuals meeting the EAR requirements for all minerals or a single mineral? If multiple minerals, which minerals were included?

Author Response

We were pleased to have the opportunity to revise our paper. We have carefully considered the comments and suggestions of you, which were helpful and constructive, and revised the paper accordingly. As instructed, we have attempted to explain the changes made in response to all of the reviewers’ comments. After addressing the issues raised, we feel the quality of the paper has greatly improved, and we hope you agree. Our responses to each comment follow herewith, and we have itemized and indicated how we have addressed each item in each comment. We have also attached a detailed response letter with the highlighted, revised sections of the manuscript. Again, thank you for the valuable and helpful comments.

Point 1: The phrase "nutrient uptake" and "uptake suitability" is used several times within this manuscript, but the use of the word uptake is improper for a study based on nutritional records since absorption is not measured. Intake (or similar terminology) should be used exclusively.

Response 1: We sincerely appreciate the reviewer’s comments. According to your comments, we used the term “intake” exclusively in the revised manuscript.

Point 2: In the Introduction, the 2021 OECD statistics should be cited.

Response 2: We have added citations to the 2021 OECD statistics accordingly. (line 55)

Added references:

  • Poverty rate(indicator). 2021; 10.1787/0fe1315d-en.

Point 3: The variable "suitability of nutrition uptake" is somewhat vaguely defined in the Materials and Methods. Is this indicating whether intake met or exceeded the EAR for a particular micronutrient or all micronutrients combined? For example, if a score of 1 is assigned, does this indicate the intake of all micronutrients exceeded the EAR? This should be stated plainly within the Methods section.

Response 3: We appreciate your comments. As your comments, we stated the definition of “suitability of nutrition uptake” more specifically in the Methods section. Specifically, assignment of a score of 1 for a particular nutrient indicates that the consumption of that specific micronutrient has exceeded the EAR and is lower than the UL. For the analysis on overall vitamins or minerals, overall intake was considered suitable when the intake of every vitamin or mineral examined was proper.

Revised manuscripts, line 93~96

The suitability of nutrition intake meant whether the intake of each target nutrient was adequate or not; 0 stood for unsuitable, and 1 stood for suitable nutrient intake.  Overall intake for vitamins and minerals was considered suitable when the intake of every vitamin or mineral examined was adequate

Point 4: It is unclear why only certain vitamins and minerals were analyzed and not all vitamins and minerals.

Response 4: Since Dietary Reference Intakes for Koreans (KDRIs) did not include EAR values for all nutrients, we decided to choose only certain vitamins and minerals for analysis. We apologize for not clearly stating the reasoning behind our decision.

Revised manuscripts, line 108~111

Among all the vitamins and minerals, vitamin A, vitamins B1/B2/B3/B9, vitamin C, cal-cium, phosphorus, and iron, whose EARs are shown in KDRIs and intake amounts are available on KNHANES, were selected as the targets of analysis.

Point 5: Appropriate significant digits should be used when displaying p-values and confidence intervals. The use of four significant digits is excessive and wholly uninformative.

Response 5: In the revised manuscript, we adjusted the significant digits of p-values from 4 to 3 according to the regulations and examples of the  journal (Tables).

Point 6: Please explain the variables "nutrition educational status" and "dietary control“

Response 6: we have added detailed explanations for the two variables accordingly.

Revised manuscripts, line 117~121

nutritional education status (whether or not to have received nutrition education and counseling conducted at public health centers, ward offices, community centers, welfare facilities, schools, or hospitals in the past year  - yes or no), dietary control (whether or not to control overall diet for special reasons such as disease, weight control - yes or no), and energy intake (kcal).

Point 7: In Table 2, why were the statistics for the other regression variables (age, weight, etc.) not performed?

Response 7: Regression analysis was performed for the age variable. Age was divided into 6 different subgroups before being included in the logistic regression model.

The regression analysis was performed with the “obesity status” variable calculated with the BMI variable, not directly with the weight variable.

We felt it was necessary to include the obesity status of individual participants in our nutritional study. Thus we utilized the obesity status variable based on BMI. Since BMI is calculated based on weight, we did not include the weight variable in our model since multicollinearity issues arise when variables are related. The criteria for determining obesity status based on BMI is stated under the corresponding tables(1,3,4).

We apologize for not clearly stating why the weight variable was not included in the regression model.

Point 8: Similar to question 3, it is unclear what is being reported in Table 4. Is this a synthesis of individuals meeting the EAR requirements for all minerals or a single mineral? If multiple minerals, which minerals were included?

Response 8: We plainly stated the definition of overall vitamin/mineral intake. For example, assigning 1 in the overall vitamin intake indicates a participant succeeded in meeting the EAR requirements and having intake below the UI across all 3 target minerals of our study.

Revised manuscripts, line 95~96

Overall intake for vitamins and minerals was considered suitable when the intake of every vitamin or mineral examined was adequate.

Additional revision:

  • The grammar in the revised manuscript has been carefully reviewed by all authors and a certified English editor.
  • A funding source has been added to the revised manuscript. This is to pay for publication fee if you allow our papers to be published in your journal. The funder had no role in the design and conduct of the study; collection, management, analysis, and interpretation of the data; preparation, review, or approval of the manuscript; and decision to submit the manuscript for publication.
  • This paper is composed of co-corresponding authors (Seung Hoon Kim & Sung-In Jang).

Reviewer 2 Report

Thank you for giving me the opportunity for review the manuscript entitled

“ The Association between Household Income Level and Suitability of Vitamin and Mineral Uptake.”

The purpose of this study was to evaluate the relationship between household income quintile and nutrient intake suitability. The topic is of interest, it is  well recognized that a diet is one of the most important lifestyle related factors that has influence on our health.

The manuscript is interesting and tables are informative, the results are thoughtfully presented,  the  manuscript in scope of the Journal however it requires some clarifications.

Lines 84-85: The authors presented:

“The variable of interest in this study was household income quintile, with quintiles 1 to 5 indicating the lowest to the highest income group.”

What are the criteria used for household income quintile? Please specify and define. What percentage of the South Korean population belongs to each quartile?

It should be also more strongly underlined that  over or underreporting of nutrients intake could occur when the data are collected by the questionnaire  and that the declared food intake may also not correspond to the nutritional status measured by laboratory means (the concentration in blood)  as a result of different bioavailability of nutrients from different food products and individual differences in metabolism. It would be useful (if available) to look at the micronutrients/vitamins level in biological samples. This should be at least discussed in the discussion section of the paper.

Author Response

We were pleased to have the opportunity to revise our paper. We have carefully considered the comments and suggestions of you, which were helpful and constructive, and revised the paper accordingly. As instructed, we have attempted to explain the changes made in response to all of the reviewers’ comments. After addressing the issues raised, we feel the quality of the paper has greatly improved, and we hope you agree. Our responses to each comment follow herewith, and we have itemized and indicated how we have addressed each item in each comment. We have also attached a detailed response letter with the highlighted, revised sections of the manuscript. Again, thank you for the valuable and helpful comments.

Point 1: Lines 84-85: The authors presented:

“The variable of interest in this study was household income quintile, with quintiles 1 to 5 indicating the lowest to the highest income group.”

What are the criteria used for household income quintile? Please specify and define. What percentage of the South Korean population belongs to each quartile?

Response 1: We specified the criteria used for defining household income quintile. The “household income quintile” is calculated based upon the value of the monthly household income divided by the square root of the number of household members. The entire participants are then divided into 5 different quintiles from the highest(5) to the lowest(1) based on the calculated value, according to their groups based on age and gender. We could not use the “monthly household income” variable by itself since a higher number of household members may cause overstatement of the monthly household income. The entire South Korean demographic is divided into 5 different household income quintiles by this definition. However, our 5088 participants are not clearly divided into 5 different groups with around 1017 participants each. We do not know the reason behind this difference, but this inconsistency was compensated for by our weighted analysis.

Revised manuscripts, line 90~92

Income was obtained as the monthly household income divided by the square root of the number of household members for every member.

Point 2: It should be also more strongly underlined that over or underreporting of nutrients intake could occur when the data are collected by the questionnaire  and that the declared food intake may also not correspond to the nutritional status measured by laboratory means (the concentration in blood)  as a result of different bioavailability of nutrients from different food products and individual differences in metabolism. It would be useful (if available) to look at the micronutrients/vitamins level in biological samples. This should be at least discussed in the discussion section of the paper.

Response 2: We appreciate the editor’s comments. We emphasized limitations of questionnaire based nutrient study due to individual differences in bioavailability and metabolism of nutrients. Since the data we used did not provide detailed mineral and vitamin levels in biological samples, we added that further laboratory analysis is required. We also compared data of our analysis and former studies based on urinary vitamin B concentration, and found consistent results. We therefore strengthened the reliability of our analysis.

Revised manuscripts, line 247~254

Estimated nutrition intake based on questionnaire responses has shown differences com-pared with their corresponding biomarkers in blood or urine samples collected through laboratory means, due to individual differences of metabolism [32]. The present study could have been implemented if the data also included accurate biomarkers measured in the biological samples of each participant. Nevertheless, we found a general consistency in results on the intake suitability of vitamins B1 and B2, which suggests that the error caused by the 24-hour dietary recall method may be insignificant.

Additional revision:

  • The grammar in the revised manuscript has been carefully reviewed by all authors and a certified English editor.
  • A funding source has been added to the revised manuscript. This is to pay for publication fee if you allow our papers to be published in your journal. The funder had no role in the design and conduct of the study; collection, management, analysis, and interpretation of the data; preparation, review, or approval of the manuscript; and decision to submit the manuscript for publication.
  • This paper is composed of co-corresponding authors (Seung Hoon Kim & Sung-In Jang).

Reviewer 3 Report

The aim of this study “ The Association between Household Income Level and Suitability of Vitamin and Mineral Uptake “, nutrients-1492119 was examine the correlation between the household income quintile and vitamin or mineral uptake suitability. Data from the 2019 KNHANES were used. The uptake suitability of overall vitamins and minerals, and of individual nutrients, were examined.

Comments

Summary: It is well structured and informative. It´s recommended to include some more results.

The introduction should be broader. In line 40 they reference articles 3-15, this is not correct. The contribution of each article should be identified individually whenever possible.

Material and methods:

A cross-sectional design is used specifically on the project "2019 KNHANES VIII-1".

Please include the participation rate.

How has the income from the students been collected? And of the people who do not work?

Has the sample size been calculated based on the population under study? Please include it.

Results:

The tables are well designed and informative.

The use of three decimal places is not suitable, please use two.

The results are well commented.

Discussion:

The first sentence does not make sense, eliminating repeating objectives is not appropriate. The main results should be presented to start the discussion.

There is a lack of bibliography with which to support its results and discuss the results.

The limitations are adequate.

Include reasons for non-participation in the discussion.

Author Response

We were pleased to have the opportunity to revise our paper. We have carefully considered the comments and suggestions of you, which were helpful and constructive, and revised the paper accordingly. As instructed, we have attempted to explain the changes made in response to all of the reviewers’ comments. After addressing the issues raised, we feel the quality of the paper has greatly improved, and we hope you agree. Our responses to each comment follow herewith, and we have itemized and indicated how we have addressed each item in each comment. We have also attached a detailed response letter with the highlighted, revised sections of the manuscript. Again, thank you for the valuable and helpful comments.

Point 1: The introduction should be broader. In line 40 they reference articles 3-15, this is not correct. The contribution of each article should be identified individually whenever possible.

Response 1: We specified the contents of different reference articles in the introduction. We stated the importance of vitamins and minerals and potential problems their deficiencies may make in the introduction.

Revised manuscripts, line 40~45

For instance, deficiency of vitamin B may cause beriberi [4], anemia [5], pellagra [6], spinal cord lesions (e.g., spina bifida) and peripheral neuropathy [7]. Severe conditions including osteoporosis, anemia, and peripheral neuropathy can also occur if an insufficient amount of minerals such as calcium, phosphorus, and iron are consumed [7-11]. Therefore, it is essential to consume proper amounts of vitamins and minerals to pre-vent such diseases.

Point 2: A cross-sectional design is used specifically on the project "2019 KNHANES VIII-1". Please include the participation rate.

Response 2: We included the participation rate of the project in the article. The overall participation rate of the project was 74.7%.

Revised manuscripts, line 70

The overall participation rate was 74.7%.

Point 3: How has the income from the students been collected? And of the people who do not work?

Response 3: The household income quintile variable was calculated even for individuals without any economic activity including students. We have added how the household income quintile variable was calculated.

Revised manuscripts, line 90~93

Income was obtained as the monthly household income divided by the square root of the number of household members for every member, including those without eco-nomic activity. Quintiles were calculated for each group, divided by age and gender.

Point 4: Has the sample size been calculated based on the population under study? Please include it. (Namely the entire population.)

Response 4: According to the knhanes 2019 user guidelines, the population number was sufficient for analysis of the whole Korean population upon utilization of weighted analysis.

Revised manuscripts, line 235~239

Last, our study was conducted based on a nationwide survey. Therefore, our sam-ple had an adequate representation of participants to draw conclusions applicable to the entire South Korean population. A weighted logistic regression analysis was used to eliminate errors that may have occurred from the sampling process, including non-response or non-participation bias.

Point 5: Appropriate significant digits should be used when displaying p-values and confidence intervals. The use of four significant digits is excessive and wholly uninformative.

Response 5: In the revised manuscript, we adjusted the significant digits of p-values from 4 to 3 according to the instructions for authors of the journal (Tables).

Point 6: (Result) The use of three decimal places is not suitable, please use two.

Response 6: The digits for tables are adjusted accordingly. Digits for p-values were adjusted from 4 to 3. The digits for AORs were adjusted from 3 to 2 (Tables).

Point 7: The first sentence does not make sense, eliminating repeating objectives is not appropriate. The main results should be presented to start the discussion.

Response 7: In the revised manuscript, we started the discussion by presenting our main results and findings. (line 194)

Point 8: There is a lack of bibliography with which to support its results and discuss the results.

Response 8: We added extra bibliography in order to support our findings and discussion accordingly. We compared the vitamin B1 and B2 intake suitability in South Korean adult populations with former studies, and found consistent results. We cited that education level has an influence on food choices in South Korea in order to support our findings that education level influences the mineral intake. We found that impact of income on health status is more pronounced in women than in men, which is consistent with our findings.

Revised manuscripts, line 226~228

In addition, the percentage of individuals consuming a suitable amount of vitamin B1 (68.3%) and vitamin B2 (66.8%) was consistent with previous reports on vitamin B1 [27] and vitamin B2 [28] in South Korean adults.

Revised manuscripts, line 268~270

In South Korea, it has been reported that the consumption of mineral and vitamin rich fruit and vegetable correlated with education levels [36].

Revised manuscripts, line 276~277

In South Korea, the impact of income and education on morbidity and self-reported health status is more pronounced in women than in men [37].

Point 9: Include reasons for non-participation in the discussion.

Response 9: We could not provide reasons for non-participation since the original data did not provide it. However, we have conducted a weighted logistic analysis in order to eliminate non-participation bias.

Revised manuscripts, line 131~133

A weighted logistic regression analysis was performed to account for the complex and stratified sampling design, and a two-sided p-value < 0.05 indicated statistical significance.

Additional revision:

  • The grammar in the revised manuscript has been carefully reviewed by all authors and a certified English editor.
  • A funding source has been added to the revised manuscript. This is to pay for publication fee if you allow our papers to be published in your journal. The funder had no role in the design and conduct of the study; collection, management, analysis, and interpretation of the data; preparation, review, or approval of the manuscript; and decision to submit the manuscript for publication.
  • This paper is composed of co-corresponding authors (Seung Hoon Kim & Sung-In Jang).

Round 2

Reviewer 3 Report

After carefully reviewing the modified manuscript, in its second version, of the article entitled "The Association between Household Income Level and Suitability of Vitamin and Mineral Uptake", and currently it´is "Association between Household Income Level and Suitability of Vitamin and Mineral intake ”(nutrients-1492119) consider that the authors have clarified the doubts and improved the presentation of the work.

I believe that this work has allowed us to identify how some nutrients are not being consumed properly in the Korean population and will allow the proposal of a nutritional improvement program in this population.